# Eigenvalues of Two-State Quantum Walks Induced by the Hadamard Walk

**DOI:** 10.3390/e22010127

**Published:** 2020-01-20

**Authors:** Shimpei Endo, Takako Endo, Takashi Komatsu, Norio Konno

**Affiliations:** 1Department of Physics, Frontier Research Institute for Interdisciplinary Science, Faculty of Science, Tohoku University, 6-3, Aoba, Aramaki-aza, Aobaku, Sendai, Miyagi 980-8578, Japan; shimpei.endo@nucl.phys.tohoku.ac.jp; 2Department of Applied Mathematics, Faculty of Engineering, Yokohama National University, Hodogaya, Yokohama 240-8501, Japan; konno@ynu.ac.jp; 3Department of Bioengineering School of Engineering, The University of Tokyo, Bunkyo, Tokyo 113-8656, Japan; komatsu@coi.t.u-tokyo.ac.jp

**Keywords:** discrete-time quantum walks, eigenvalues, localization, stationary measure, classification

## Abstract

Existence of the eigenvalues of the discrete-time quantum walks is deeply related to localization of the walks. We revealed, for the first time, the distributions of the eigenvalues given by the splitted generating function method (the SGF method) of the space-inhomogeneous quantum walks in one dimension we had treated in our previous studies. Especially, we clarified the characteristic parameter dependence for the distributions of the eigenvalues with the aid of numerical simulation.

## 1. Introduction

The discrete-time quantum walks (DTQWs), as quantum counterparts of the classical random walks that play important roles in various fields, have attracted much attention in the past two decades [1,2,3,4,5,6,7]. As the reviews of the DTQWs, the readers may be referred to the work in [5,7], for instance. One of the characteristic properties of the DTQWs is localization, which is defined that the probability a walker is found at a point does not converge to zero even in the long-time limit. It has been known that there are two-state QWs in one dimension that have localization [8,9,10,11].

Localization of the DTQWs is closely related to the existence of the eigenvalues of the unitary transition operators. However, there were few results for the study of localization from the viewpoint of the eigenvalues of the unitary operators, though there were many approaches for localization [8,9,11,12,13]. As the rare results, Komatsu and Konno [14] revealed the continuous part of the spectrum of the Hadamard walk, which has attracted much attention for a decade [5,7,11,15] and showed that the Hadamard walk does not have localization. They also cleared the area of bounded-type stationary measures, stationary measures with quadratic divergence, and with exponential divergence.

In this paper, we focus on the space-inhomogeneous QWs in one dimension. As simple cases, we treat the QWs induced by the Hadamard walk we had studied in our previous studies [8,16,17,18,19]. So far, it has not been clarified the influence of defects to the distributions of the eigenvalues for the eigenfunctions in $l^{2}$-space on Z of our models, where Z is the set of integers. Also, the parameter dependence of the eigenvalues has not been known. We consider those issues with the help of numerical simulation. Our study may help to discuss localization, lead to classify the stationary measures, and construct a relation with the Spectrum scattering theory. Here, one of the significance to study the stationary measures is to clear the correspondence with that of the classical systems, i.e., the Markov chains.

The rest of this paper is organized as follows. The definitions of our DTQWs and the main results are given in Section 2. The remaining section is devoted to summarize our results.

## 2. Definitions of the DTQWs and the Main Results

In this paper, we consider the DTQWs on Z. The quantum walker with two coin states |L〉 and |R〉 is supposed to locate at each lattice point on Z by superposition. The system is described on a tensor Hilbert space $H_{p}⊗H_{C}$. The Hilbert space $H_{p}$ alters the positions and is spanned by the orthogonal normalized basis {|x〉;x∈Z}. Also, the Hilbert space $H_{C}$ represents the coin states and is spanned by the orthogonal normalized basis {|J〉:J=L,R}. We are here allowed to define
$$|L\rangle =\left[\begin{array}{c} 1\\ 0 \end{array}\right],\;|R\rangle =\left[\begin{array}{c} 0\\ 1 \end{array}\right],$$
for instance. We call |L〉 and |R〉, the left and right chiralities, respectively.

The DTQWs are defined as unitary processes in which each coin state at each position varies with given unitary operations. The quantum walker in this paper is also manipulated by unitary operations. The system of the DTQW at time *t* is represented by
$$|\Psi_{t}\rangle =[\cdots ,|\Psi_{t}\left(-2\right)\rangle ,|\Psi_{t}\left(-1\right)\rangle ,|\Psi_{t}\left(0\right)\rangle ,|\Psi_{t}\left(1\right)\rangle ,|\Psi_{t}\left(2\right)\rangle ,\cdots ]{}^{T}\;\in H_{p}⊗H_{C},$$
where $|\Psi_{t}\left(x\right)\rangle =\left[\Psi_{t}^{L}\left(x\right),\Psi_{t}^{R}\left(x\right)\right]{}^{T}\;$ is the amplitude of the DTQW at time *t*. Here, *L* and *R* correspond to the left and right chirarities, respectively, and *T* stands for the transposed operator. Let us prepare a sequences of 2×2 unitary matrices $A=\{A_{x}:x\in Z\}$ with
$$\begin{array}{c} A_{x}=\left[\begin{array}{cc} a_{x} & b_{x}\\ c_{x} & d_{x} \end{array}\right]. \end{array}$$

Put
$$U^{\left(s\right)}=S{⊕}_{x\in Z}A_{x},$$
where *S* is the standard shift operator defined by
$$S=\sum_{x}(|x\rangle \langle x+1|⊗|L\rangle \langle L|+(|x\rangle \langle x-1|⊗|R\rangle \langle R|).$$

In our study, we consider the eigenvalue problem
$$U^{\left(s\right)}\Psi =\lambda \Psi \left(\Psi \in Map\left(Z,C^{2}\right),\lambda \in S^{1}\right),$$
where $S^{1}=\left\{z\in C;|z|=1\right\}.$ The SGF method developed in [11] is one of the effective methods to solve the eigenvalue problem and construct the stationary measure by using the generating functions of the amplitudes of the QW. The details are described in [17], for instance.

The time evolution is determined by
$$\begin{array}{c} \Psi_{t+1}\left(x\right)=\left(U^{\left(s\right)}\Psi_{t}\right)\left(x\right)=P_{x+1}\Psi_{t}\left(x+1\right)+Q_{x-1}\Psi_{t}\left(x-1\right)\;\left(x\in Z\right), \end{array}$$
where
$$\begin{array}{c} P_{x}=\left[\begin{array}{cc} a_{x} & b_{x}\\ 0 & 0 \end{array}\right],\;Q_{x}=\left[\begin{array}{cc} 0 & 0\\ c_{x} & d_{x} \end{array}\right] \end{array}$$
with $A_{x}=P_{x}+Q_{x}$. Note that $P_{x}$ and $Q_{x}$ express the left and right movements, respectively (Figure 1).

Hereafter, as simple cases of the space-inhomogeneous QWs in one dimension, we focus on the QWs originated from the Hadamard walk we had treated in our previous studies [8,16,17,18,19]. In those studies, we obtained the spectrum for the generalized eigenfunctions and corresponding stationary measure for each QW model, and found that there are eigenvalues that give the stationary measure with exponential decay. Motivated by those studies and the work in [14,20], we focus on the eigenvalues movements in accordance with model parameters. In particular, we investigate the eigenvalues movements for the QWs with one defect and the two-phase QWs that steps differently in positive and negative parts.

### 2.1. Model 1: The Wojcik Model

At first, we focus on the Wojcik model, whose unitary transition operators are (Figure 2)
(1)$$\begin{array}{c} {\left\{A_{x}\right\}}_{x\in Z}=\left\{\omega H_{x=0},\;H_{x\in Z\setminus \{0\}}\right\} \end{array}$$
with $\omega =e^{2i\pi \phi },$ where ϕ∈(0,1). Here *H* is the Hadamard gate defined by
$$\begin{array}{c} H=\frac{1}{\sqrt{2}}\left[\begin{array}{cc} 1 & 1\\ 1 & -1 \end{array}\right]. \end{array}$$

The Hadamard walk can be given by ϕ→0 in Equation (Equation 1). We note that the Wojcik model has a phase 2πϕ only at the origin. By using recurrence equations, Wojcik et al. [21] solved the eigenvalue problem. Endo et al. [17] and Endo and Konno [18] derived the stationary, the time-averaged limit, and the weak limit measures. They discussed localization and weak convergence, respectively.

Now let $\Psi \left(x\right)=\left[\Psi^{L}\left(x\right),\Psi^{R}\left(x\right)\right]{}^{T}\;$ be the amplitude, and put $\alpha =\Psi^{L}\left(0\right)$ and $\beta =\Psi^{R}\left(0\right)$. Endo et al. [17] solved the eigenvalue problem
$$U^{\left(s\right)}\Psi =\lambda \Psi \left(\Psi \in Map\left(Z,C^{2}\right),\lambda \in S^{1}\right).$$

Here, we give the illustrations of the movements of the eigenvalues given by the SGF method [11] of the Wojcik model (Figure 3) and the table of the parameter dependence of the eigenvalues (Table 1). We remark that each illustration is a diagram of the numerical simulation to investigate the parameter dependence continuously by using mathematica. The SGF method gives the stationary measures with exponential and constant cases.

Note that the eigenvalues can be obtained by Equations (3.8) and (3.9) of Proposition 1 in [17]:
(1)β=iα case.
$$\begin{array}{c} \lambda^{2}=\frac{\omega \left(1-2\omega +\omega^{2}\right)-i\omega \left(1-\omega +\omega^{2}\right)}{1-2\omega +2\omega^{2}}. \end{array}$$(2)β=−iα case.
$$\begin{array}{c} \lambda^{2}=\frac{\omega \left(1-2\omega +\omega^{2}\right)+i\omega \left(1-\omega +\omega^{2}\right)}{1-2\omega +2\omega^{2}}. \end{array}$$
Letting
$$\left\{\begin{array}{cc} \lambda^{\left(1\right)}\left(\phi \right):=\sqrt{\lambda^{2}},\lambda^{\left(2\right)}\left(\phi \right):=-\sqrt{\lambda^{2}} & (\beta =i\alpha ),\\ \lambda^{\left(3\right)}\left(\phi \right):=\sqrt{\lambda^{2}},\lambda^{\left(4\right)}\left(\phi \right):=-\sqrt{\lambda^{2}} & (\beta =-i\alpha ), \end{array}\right.$$
we specified the regions of the parameter ϕ that lead to the eigenfunctions in $l^{2}$-space on Z by elementary analytic calculations, that is, we have $\phi \in (\frac{1}{4},1)$ for β=iα case, and $\phi \in (0,\frac{3}{4})$ for β=−iα case.

Now let σ(H) be the region of the continuous spectrum of the Hadamard walk. We see that the Wojcik model does not have the eigenvalues on σ(H) in the range of the parameter ϕ. We also notice that despite the two divided cases of the initial state, the distributions of the eigenvalues are the same, and the eigenvalues move allover $S^{1}\setminus \sigma \left(H\right)$.

### 2.2. Model 2: The Hadamard Walk with One Defect

Next, our one-defect model is defined by the set of unitary matrices (Figure 3)
(2)$$\begin{array}{c} {\left\{A_{x}\right\}}_{x\in Z}=\left\{{\left[\begin{array}{cc} \cos\xi & \sin\xi \\ \sin\xi & -\cos\xi \end{array}\right]}_{x=0},\;H_{x\in Z\setminus \{0\}}\right\} \end{array}$$
with ξ∈(0,π/2). We can extend some cases to ξ=0 or ξ=π/2.

Put $\alpha =\Psi^{L}\left(0\right)$ and $\beta =\Psi^{R}\left(0\right)$. Then, the solutions of the eigenvalue problem
$$\begin{array}{c} U^{\left(s\right)}\Psi =\lambda \Psi \left(\Psi \in Map\left(Z,C^{2}\right),\lambda \in S^{1}\right) \end{array}$$
are given in [8].

Now we show the illustrations of the movements of the eigenvalues given by the SGF method of the Hadamard walk with one defect (Figure 4) and the table of the parameter dependence of the eigenvalues (Table 2). We remark that each illustration is a diagram of the numerical simulation to investigate the parameter dependence continuously by using mathematica.

Note that the eigenvalues can be obtained by Proposition 3.1 in [8]:

Put C=cosξ and S=sinξ.
(1)β=−iα case. We get
$$\begin{array}{c} \lambda =\pm \frac{C+(\sqrt{2}-S)i}{\sqrt{3-2\sqrt{2}S}}. \end{array}$$(2)β=iα case. We get
$$\begin{array}{c} \lambda =\pm \frac{C-(\sqrt{2}-S)i}{\sqrt{3-2\sqrt{2}S}}. \end{array}$$
Putting
$$\left\{\begin{array}{cc} \lambda^{\left(1\right)}\left(\xi \right):=\sqrt{\lambda^{2}},\lambda^{\left(2\right)}\left(\xi \right):=-\sqrt{\lambda^{2}} & (\beta =-i\alpha ),\\ \lambda^{\left(3\right)}\left(\xi \right):=\sqrt{\lambda^{2}},\lambda^{\left(4\right)}\left(\xi \right):=-\sqrt{\lambda^{2}} & (\beta =i\alpha ), \end{array}\right.$$
we found the regions of the parameter ξ that connect to the eigenfunctions in $l^{2}$-space on Z by basic analytic calculations, i.e., $\xi \in (o,\frac{\pi }{4})$.

We emphasize that the eigenvalues emerge only on $S^{1}\setminus \sigma \left(H\right)$. We also notice that the eigenvalues turn in the opposite direction for the two divided cases of the initial state, and the movements of the eigenvalues don’t cover $S^{1}\setminus \sigma \left(H\right)$.

### 2.3. Model 3: The Two-Phase QW with One Defect

Here we consider the QW whose time-evolution is determined by the unitary transition operators (Figure 5)
(3)$$\begin{array}{c} {\left\{A_{x}\right\}}_{x\in Z}=\left\{{\left[\begin{array}{cc} 1 & 0\\ 0 & -1 \end{array}\right]}_{x=0},\;\frac{1}{\sqrt{2}}{\left[\begin{array}{cc} 1 & e^{i\sigma_{+}}\\ e^{-i\sigma_{+}} & -1 \end{array}\right]}_{x\geq 1},\;\frac{1}{\sqrt{2}}{\left[\begin{array}{cc} 1 & e^{i\sigma_{-}}\\ e^{-i\sigma_{-}} & -1 \end{array}\right]}_{x\leq -1}\right\} \end{array}$$
with $\sigma_{\pm }\in R$. The quantum walker shifts differently in positive and negative parts respectively, and the determinants are independent of the position, that is, $\det(U_{x})=-1$ for x∈Z. The model is called “the two-phase QW with one defect” for short. If $\sigma_{+}=\sigma_{-}$, the model becomes a one-defect QW which has been so far analyzed in detail [11]. We should notice that our model has a defect at the origin, which enables us to analyze the model simply.

Now the solutions of the eigenvalue problem
$$U^{\left(s\right)}\Psi =\lambda \Psi \left(\Psi \in Map\left(Z,C^{2}\right),\lambda \in S^{1}\right)$$
are described in [9]. Here, we put on the illustration of the movements of the eigenvalues given by the SGF method of the two-phase QW with one defect (Figure 6) and the table of the parameter dependence of the eigenvalues (Table 3).

We remark that the illustration is a diagram of the numerical simulation to investigate the parameter dependence continuously by using mathematica. Note that the eigenvalues can be obtained by Proposition 1 in [18]:

By putting $\sigma =(\sigma_{+}-\sigma_{-})/2$, we have
$\lambda^{\left(1\right)}=\frac{\cos\sigma +(\sin\sigma +\sqrt{2})i}{\sqrt{3+2\sqrt{2}\sin\sigma }},\;\lambda^{\left(2\right)}=-\lambda^{\left(1\right)}.$$\lambda^{\left(3\right)}=-\frac{\cos\sigma +(\sin\sigma -\sqrt{2})i}{\sqrt{3-2\sqrt{2}\sin\sigma }},\;\lambda^{\left(4\right)}=-\lambda^{\left(3\right)}.$
Letting
$$\lambda^{\left(1\right)}\left(\sigma \right):=\lambda^{\left(1\right)},\lambda^{\left(2\right)}\left(\sigma \right):=\lambda^{\left(2\right)},\lambda^{\left(3\right)}\left(\sigma \right):=\lambda^{\left(3\right)},\lambda^{\left(4\right)}\left(\sigma \right):=\lambda^{\left(4\right)},$$
we specified the regions of the parameter σ that lead to the eigenfunctions in $l^{2}$-space on Z by elementary analytic calculations, that is, we have $\sigma \in \left[0,\frac{5}{4}\pi \right)\cup \left(\frac{7}{4}\pi ,2\pi \right]$ for $\lambda^{\left(1\right)}\left(\sigma \right)$ and $\lambda^{\left(2\right)}\left(\sigma \right),$ and $\sigma \in \left[0,\frac{1}{4}\pi \right)\cup \left(\frac{3}{4}\pi ,2\pi \right]$ for $\lambda^{\left(3\right)}\left(\sigma \right)$ and $\lambda^{\left(4\right)}\left(\sigma \right).$

We see that the two-phase quantum walk with one defect does not have the eigenvalues on σ(H) in the range of the parameter ϕ. (Figure 7) We also notice that $\lambda^{1}\left(\sigma \right)$ and $\lambda^{3}\left(\sigma \right)$, $\lambda^{2}\left(\sigma \right)$ and $\lambda^{4}\left(\sigma \right)$ turn in the same direction, respectively, and the eigenvalues move allover $S^{1}\setminus \sigma \left(H\right)$.

### 2.4. Model 4: The Complete Two-phase QW

Lastly, we introduce the QW which does not have defects, whose unitary matrices are (Figure 8)
(4)$$\begin{array}{c} {\left\{A_{x}\right\}}_{x\in Z}=\left\{\frac{1}{\sqrt{2}}{\left[\begin{array}{cc} 1 & e^{i\sigma_{+}}\\ e^{-i\sigma_{+}} & -1 \end{array}\right]}_{x\geq 0},\;\frac{1}{\sqrt{2}}{\left[\begin{array}{cc} 1 & e^{i\sigma_{-}}\\ e^{-i\sigma_{-}} & -1 \end{array}\right]}_{x\leq -1}\right\} \end{array}$$
with $\sigma_{\pm }\in R$. The walker steps differently in the spatial regions x≥0 and x≤−1 with the phase parameters $\sigma_{+}$ and $\sigma_{-}$. The QW does not have defect at the origin, which is in marked contrast to the two-phase QW with one defect [9,16]. Hereafter, we call the QW the complete two-phase QW. Putting $\sigma_{+}=\sigma_{-}=0$, the model becomes the Hadamard walk studied in [8,11,22].

Let us consider the eigenvalue problem
$$U^{\left(s\right)}\Psi =\lambda \Psi \left(\Psi \in Map\left(Z,C^{2}\right),\lambda \in S^{1}\right),$$
whose solutions are given in [16]. Now we show the illustration of the movements of the eigenvalues given by the SGF method of the complete two-phase QW (Figure 9) and the table of the parameter dependence of the eigenvalues (Table 4). We remark that each illustration is a diagram of the numerical simulation to investigate the parameter dependence continuously by using mathematica.

Note that the eigenvalues can be obtained by Theorem 1 in [16]:

Let $\lambda^{\left(j\right)}$ be the eigenvalues of the unitary matrix $U^{\left(s\right)}$, and $\Psi^{\left(j\right)}\left(0\right)$ be the eigenvector at x=0, with j=1,2,3,4. Put
$$\begin{array}{c} \left\{\begin{array}{c} p=e^{i\sigma_{+}}\left(e^{-2i\sigma_{-}}-e^{-2i\sigma_{+}}-4e^{-2i\tilde{\sigma }}\right),\\ q=e^{-2i\sigma_{-}}+e^{-2i\sigma_{+}}-6e^{-2i\tilde{\sigma }},\\ r^{\left(\pm \right)}=e^{-i\sigma_{+}}\pm e^{-i\sigma_{-}}, \end{array}\right. \end{array}$$
where $\tilde{\sigma }=\left(\sigma_{+}+\sigma_{-}\right)/2$ and $c\in R_{+}$ with $R_{+}=\left(0,\infty \right)$. Then, we have
$\lambda^{\left(1\right)}=\sqrt{\frac{p+e^{i\sigma_{+}}r^{\left(-\right)}\sqrt{q}}{2(-r^{\left(-\right)}-\sqrt{q})}},\;\lambda^{\left(2\right)}=-\lambda^{\left(1\right)}.$$\lambda^{\left(3\right)}=\sqrt{\frac{p-e^{i\sigma_{+}}r^{\left(-\right)}\sqrt{q}}{2(-r^{\left(-\right)}+\sqrt{q})}},\;\lambda^{\left(4\right)}=-\lambda^{\left(3\right)}.$
Putting
$$\lambda^{\left(1\right)}\left(\sigma \right):=\lambda^{\left(1\right)},\lambda^{\left(2\right)}\left(\sigma \right):=\lambda^{\left(2\right)},\lambda^{\left(3\right)}\left(\sigma \right):=\lambda^{\left(3\right)},\lambda^{\left(4\right)}\left(\sigma \right):=\lambda^{\left(4\right)},$$
we specified the regions of the parameter σ that connect to the eigenfunctions in $l^{2}$-space on Z by basic analytic calculations, that is, we have $\sigma \in \left[\frac{1}{2}\pi ,\pi \right)\cup \left(\frac{3}{2}\pi ,2\pi \right]$ for $\lambda^{\left(1\right)}\left(\sigma \right)$ and $\lambda^{\left(2\right)}\left(\sigma \right),$ and $\sigma \in \left[0,\frac{1}{2}\pi \right)\cup \left(\pi ,\frac{3}{2}\pi \right]$ for $\lambda^{\left(3\right)}\left(\sigma \right)$ and $\lambda^{\left(4\right)}\left(\sigma \right).$

We emphasize that the eigenvalues emerge only on $S^{1}\setminus \sigma \left(H\right)$. We also notice that each eigenvalue turns in a orbit two times, and the movements of the eigenvalues cover allover $S^{1}\setminus \sigma \left(H\right)$.

## 3. Summary

As simple cases, we focused on four kinds of the space-inhomogeneous QW models in one dimension originated from the Hadamard walk. According to the previous studies [8,16,17,18,19], our QW models have the eigenvalues for the eigenfunctions in $l^{2}$-space on Z, i.e., localization occur [23,24,25,26]; however, we could not see where the eigenvalues arise in the unit circle. As a result, we clarified for the first time the characteristic distributions of the eigenvalues given by the SGF method. Specifically, we revealed that the eigenvalues do not emerge on the region of the continuous spectrum of the Hadamard walk, which implies that the continuous spectrum of our QW models may coincide with that of the Hadamard walk, consistent with the authors of [20] who claim that the continuous spectrum does not change with a finite number of defects. By using mathematica, we cleared continuously the parameter dependence of our models on the eigenvalues. One of the basic future problems is to generalize our results to the one dimensional QW models.

## Figures and Tables

**Figure 1 entropy-22-00127-f001:**
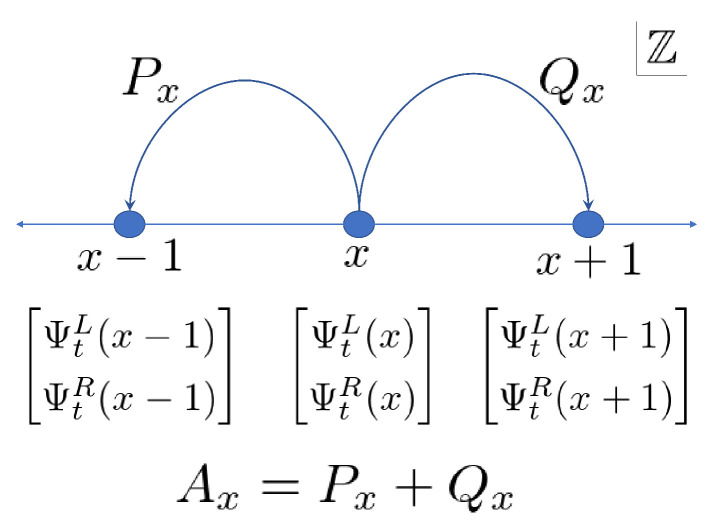
Pattern diagram of the QW.

**Figure 2 entropy-22-00127-f002:**
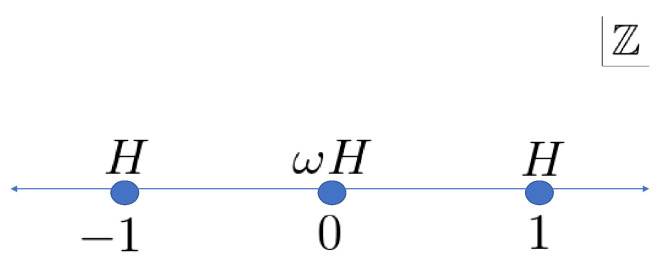
Arrangement of the unitary operators of the Wojcik model.

**Figure 3 entropy-22-00127-f003:**
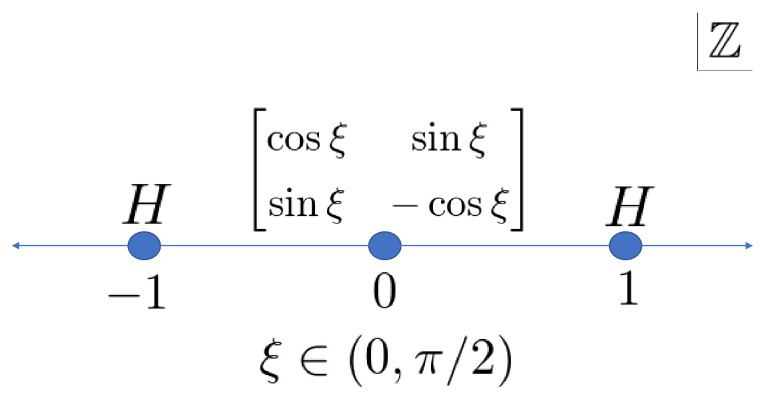
Arrangement of the unitary operators of the Hadamard walk with one defect.

**Figure 4 entropy-22-00127-f004:**
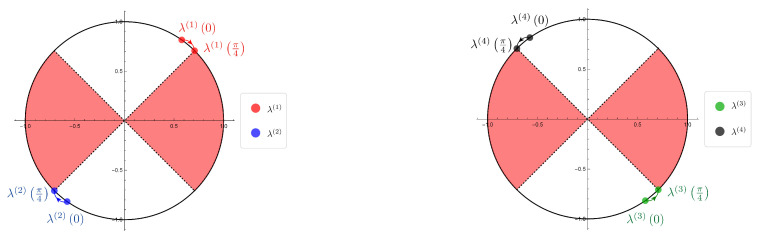
The illustrations of the eigenvalues movements of the Hadamard walk with one defect (1) β=−iα case. $\lambda^{\left(1\right)}\left(\xi \right),\;\lambda^{\left(2\right)}\left(\xi \right)\;\left(\xi \in \left(0,\frac{\pi }{4}\right)\right)$
(2) β=iα case. $\lambda^{\left(3\right)}\left(\xi \right),\;\lambda^{\left(4\right)}\left(\xi \right)\;\left(\xi \in \left(0,\frac{\pi }{4}\right)\right)$ (The red part is the region of the continuous spectrum of the Hadamard walk, i.e., σ(H)).

**Figure 5 entropy-22-00127-f005:**
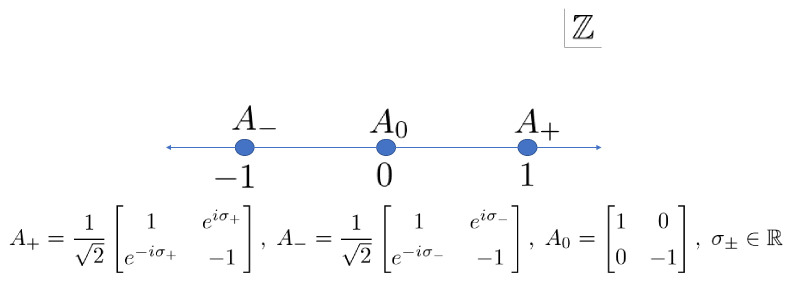
Arrangement of the unitary operators of the two-phase QW with one defect.

**Figure 6 entropy-22-00127-f006:**
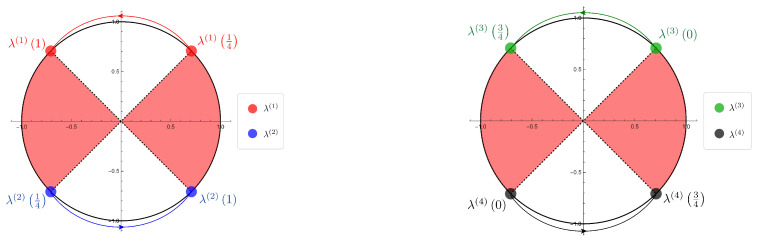
The illustrations of the eigenvalues movements of the Wojcik model (1) β=iα case. $\lambda^{\left(1\right)}\left(\phi \right),\;\lambda^{\left(2\right)}\left(\phi \right)\;\left(\phi \in \left(\frac{1}{4},1\right)\right)$
(2) β=−iα case. $\lambda^{\left(3\right)}\left(\phi \right),\;\lambda^{\left(4\right)}\left(\phi \right)\;\left(\phi \in \left(0,\frac{3}{4}\right)\right)$ (The red part is the region of the continuous spectrum of the Hadamard walk, i.e., σ(H)).

**Figure 7 entropy-22-00127-f007:**
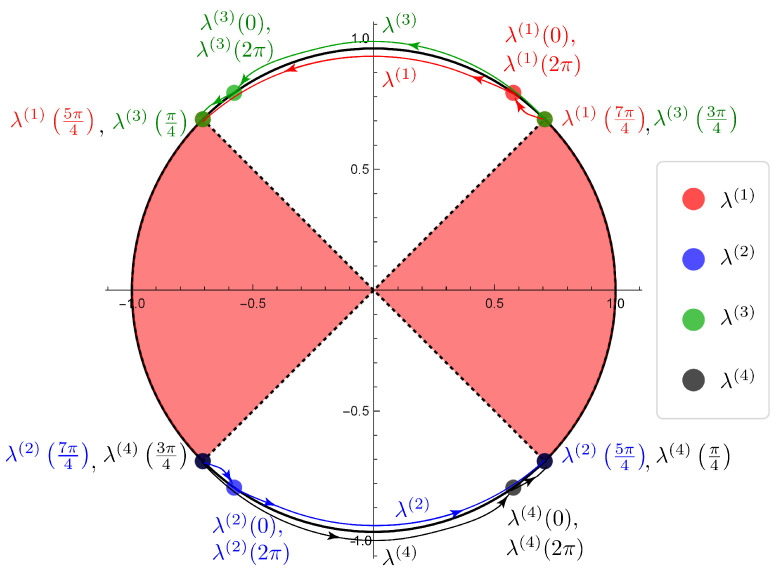
The illustration of the eigenvalues movements of the two-phase quantum walk with one defect For $\lambda^{\left(1\right)}\left(\sigma \right)$ and $\lambda^{\left(2\right)}\left(\sigma \right),$ we have $\sigma \in \left[0,\frac{5}{4}\pi \right)\cup \left(\frac{7}{4}\pi ,2\pi \right].$ For $\lambda^{\left(3\right)}\left(\sigma \right)$ and $\lambda^{\left(4\right)}\left(\sigma \right),$ we have $\sigma \in \left[0,\frac{1}{4}\pi \right)\cup \left(\frac{3}{4}\pi ,2\pi \right].$ (The red part is the region of the continuous spectrum of the Hadamard walk, i.e., σ(H)).

**Figure 8 entropy-22-00127-f008:**
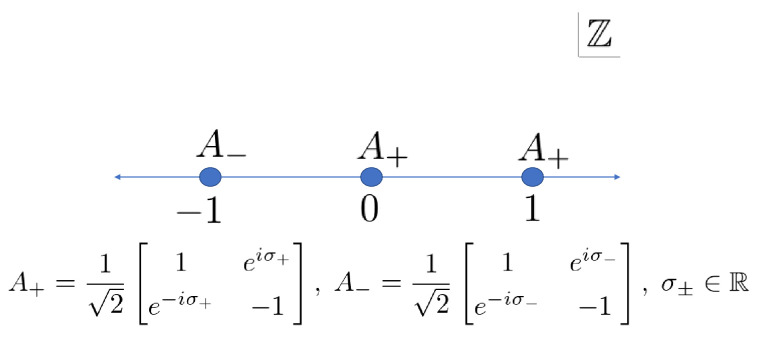
Arrangement of the unitary operators of the two-phase QW with one defect.

**Figure 9 entropy-22-00127-f009:**
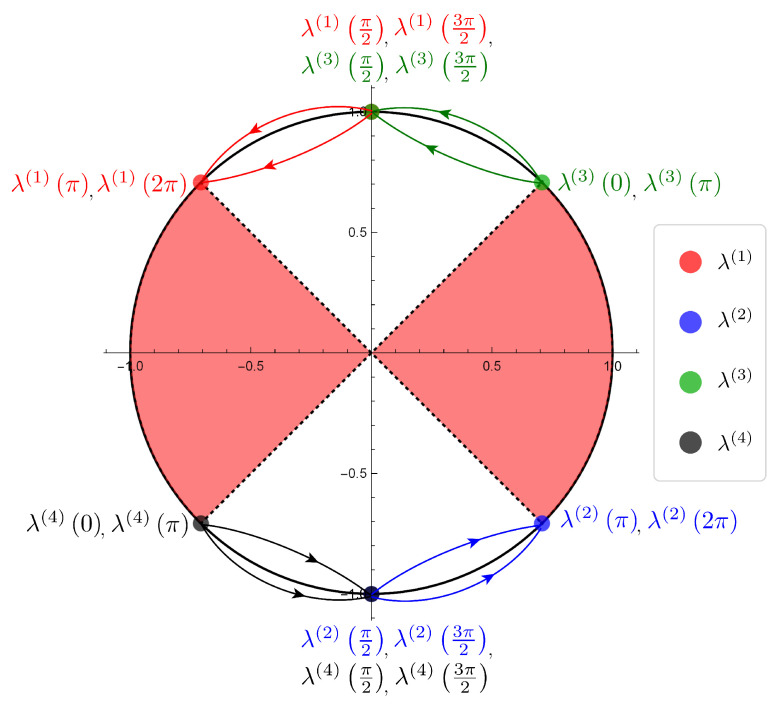
The illustration of the eigenvalues movements of the complete two-phase quantum walk For $\lambda^{\left(1\right)}\left(\sigma \right)$ and $\lambda^{\left(2\right)}\left(\sigma \right),$ we have $\sigma \in \left[\frac{1}{2}\pi ,\pi \right)\cup \left(\frac{3}{2}\pi ,2\pi \right].$ For $\lambda^{\left(3\right)}\left(\sigma \right)$ and $\lambda^{\left(4\right)}\left(\sigma \right),$ we have $\sigma \in \left[0,\frac{1}{2}\pi \right)\cup \left(\pi ,\frac{3}{2}\pi \right].$ (The red part is the region of the continuous spectrum of the Hadamard walk, i.e., σ(H)).

**Table 1 entropy-22-00127-t001:** The parameter dependence of the eigenvalues and the corresponding stationary measures for the Wojcik model.

Value of ϕ	(1)β=iα Case
$\frac{1}{8}\;\left(\omega =e^{\frac{\pi }{4}i}\right)$	$\lambda^{2}=-1,\;\mu \left(x\right)=2{\left|\alpha \right|}^{2}{\left(\frac{1}{3-2\sqrt{2}}\right)}^{\left|x\right|}\times \left\{\begin{array}{c} 2-\sqrt{2}\;\left(x\neq 0\right)\\ 1\;(x=0) \end{array}\right.$
$\frac{1}{6}\;\left(\omega =e^{\frac{\pi }{3}i}\right)$	$\lambda^{2}=-\frac{1}{2}+\frac{\sqrt{3}}{2}i,\;\mu \left(x\right)=2{\left|\alpha \right|}^{2}{\left(\frac{1}{2-\sqrt{3}}\right)}^{\left|x\right|}\times \left\{\begin{array}{c} \frac{3}{2}-\frac{\sqrt{3}}{2}\;\left(x\neq 0\right)\\ 1\;(x=0) \end{array}\right.$
$\frac{1}{4}\;\left(\omega =e^{\frac{\pi }{2}i}\right)$	$\lambda^{2}=i,\;\mu \left(x\right)=2{\left|\alpha \right|}^{2}$
$\frac{1}{2}\;\left(\omega =i\right)$	$\lambda^{2}=-\frac{4}{5}+\frac{3}{5}i,\;\mu \left(x\right)=2{\left|\alpha \right|}^{2}{\left(\frac{1}{5}\right)}^{\left|x\right|}\times \left\{\begin{array}{c} 3\;(x\neq 0)\\ 1\;(x=0) \end{array}\right.$
$\frac{2}{3}\;\left(\omega =e^{\frac{4\pi }{3}i}\right)$	$\lambda^{2}=\frac{-15-6\sqrt{3}-\left(9\sqrt{3}-10\right)i}{26},\;\mu \left(x\right)=2{\left|\alpha \right|}^{2}{\left(\frac{1}{4+\sqrt{3}}\right)}^{\left|x\right|}\times \left\{\begin{array}{c} \frac{5}{2}+\frac{\sqrt{3}}{2}\;\left(x\neq 0\right)\\ 1\;(x=0) \end{array}\right.$
$\frac{3}{4}\;\left(\omega =e^{\frac{3\pi }{2}i}\right)$	$\lambda^{2}=-\frac{4}{5}-\frac{3}{5}i,\;\mu \left(x\right)=2{\left|\alpha \right|}^{2}{\left(\frac{1}{5}\right)}^{\left|x\right|}\times \left\{\begin{array}{c} 3\;(x\neq 0)\\ 1\;(x=0) \end{array}\right.$
**Value of** ϕ	(2)β=−iα **Case**
$\frac{1}{8}\;\left(\omega =e^{\frac{\pi }{4}i}\right)$	$\lambda^{2}=-1,\;\mu \left(x\right)=2{\left|\alpha \right|}^{2}{\left(\frac{1}{3}\right)}^{\left|x\right|}\times \left\{\begin{array}{c} 2\;(x\neq 0)\\ 1\;(x=0) \end{array}\right.$
$\frac{1}{6}\;\left(\omega =e^{\frac{\pi }{3}i}\right)$	$\lambda^{2}=-\frac{1}{2}-\frac{\sqrt{3}}{2}i,\;\mu \left(x\right)=2{\left|\alpha \right|}^{2}{\left(\frac{1}{2-\sqrt{3}}\right)}^{\left|x\right|}\times \left\{\begin{array}{c} \frac{3}{2}-\frac{\sqrt{3}}{2}\;\left(x\neq 0\right)\\ 1\;(x=0) \end{array}\right.$
$\frac{1}{4}\;\left(\omega =e^{\frac{\pi }{2}i}\right)$	$\lambda^{2}=-i,\;\mu \left(x\right)=2{\left|\alpha \right|}^{2}{\left(\frac{1}{5}\right)}^{\left|x\right|}\times \left\{\begin{array}{c} 3\;(x\neq 0)\\ 1\;(x=0) \end{array}\right.$
$\frac{1}{2}\;\left(\omega =i\right)$	$\lambda^{2}=-\frac{4}{5}-\frac{3}{5}i,\;\mu \left(x\right)=2{\left|\alpha \right|}^{2}{\left(\frac{1}{5}\right)}^{\left|x\right|}\times \left\{\begin{array}{c} 3\;(x\neq 0)\\ 1\;(x=0) \end{array}\right.$
$\frac{2}{3}\;\left(\omega =e^{\frac{4\pi }{3}i}\right)$	$\lambda^{2}=\frac{-15-6\sqrt{3}+\left(9\sqrt{3}-10\right)i}{26},\;\mu \left(x\right)=2{\left|\alpha \right|}^{2}{\left(\frac{1}{4+\sqrt{3}}\right)}^{\left|x\right|}\times \left\{\begin{array}{c} \frac{5}{2}+\frac{\sqrt{3}}{2}\;\left(x\neq 0\right)\\ 1\;(x=0) \end{array}\right.$
$\frac{3}{4}\;\left(\omega =e^{\frac{3\pi }{2}i}\right)$	$\lambda^{2}=-\frac{4}{5}+\frac{3}{5}i,\;\mu \left(x\right)=2{\left|\alpha \right|}^{2}$

**Table 2 entropy-22-00127-t002:** The parameter dependence of the eigenvalues and the corresponding stationary measures for the Hadamard walk with one defect.

Value of ξ	(1)β=−iα Case
ξ=0	$\lambda =\pm \frac{1+\sqrt{2}i}{\sqrt{3}},\;\mu \left(x\right)={\left|c\right|}^{2}{\left(\frac{1}{3}\right)}^{\left|x\right|}\times \left\{\begin{array}{c} 2\;(x\neq 0)\\ 1\;(x=0) \end{array}\right.$
$\xi =\frac{\pi }{4}$	$\lambda =\pm \frac{1}{\sqrt{2}}\left(1+i\right),\;\mu \left(x\right)={\left|c\right|}^{2}$
$\xi =\frac{\pi }{2}$	$\lambda =\pm i,\;,\;\mu \left(x\right)={\left|c\right|}^{2}{\left(\frac{1}{3-2\sqrt{2}}\right)}^{\left|x\right|}\times \left\{\begin{array}{c} 2-\sqrt{2}\;\left(x\neq 0\right)\\ 1\;(x=0) \end{array}\right.$
**Value of** ξ	(2)β=iα **Case**
ξ=0	$\lambda =\pm \frac{1-\sqrt{2}i}{\sqrt{3}},\;\mu \left(x\right)={\left|c\right|}^{2}{\left(\frac{1}{3}\right)}^{\left|x\right|}\times \left\{\begin{array}{c} 2\;(x\neq 0)\\ 1\;(x=0) \end{array}\right.$
$\xi =\frac{\pi }{4}$	$\lambda =\pm \frac{1}{\sqrt{2}}\left(1-i\right),\;\mu \left(x\right)={\left|c\right|}^{2}$
$\xi =\frac{\pi }{2}$	$\lambda =\pm i,\;,\;\mu \left(x\right)={\left|c\right|}^{2}{\left(\frac{1}{3-2\sqrt{2}}\right)}^{\left|x\right|}\times \left\{\begin{array}{c} 2-\sqrt{2}\;\left(x\neq 0\right)\\ 1\;(x=0) \end{array}\right.$

**Table 3 entropy-22-00127-t003:** The parameter dependence of the eigenvalues for the two-phase quantum walk with one defect.

Value of σ	The Eigenvalues
σ=0	$\lambda^{\left(1\right)}=\frac{1+\sqrt{2}i}{\sqrt{3}},\;\lambda^{\left(2\right)}=-\lambda^{\left(1\right)},\;\lambda^{\left(3\right)}=\frac{-1+\sqrt{2}i}{\sqrt{3}},\;\lambda^{\left(4\right)}=-\lambda^{\left(3\right)}$
$\sigma =\frac{\pi }{4}$	$\lambda^{\left(1\right)}=\frac{1+3i}{\sqrt{10}},\;\lambda^{\left(2\right)}=-\lambda^{\left(1\right)},\;\lambda^{\left(3\right)}=\frac{-1+i}{\sqrt{2}},\;\lambda^{\left(4\right)}=-\lambda^{\left(3\right)}$
$\sigma =\frac{\pi }{2}$	$\lambda^{\left(1\right)}=i,\;\lambda^{\left(2\right)}=-\lambda^{\left(1\right)},\;\lambda^{\left(3\right)}=i,\;\lambda^{\left(4\right)}=-\lambda^{\left(3\right)}$
$\sigma =\frac{3}{4}\pi$	$\lambda^{\left(1\right)}=\frac{-1+3i}{\sqrt{10}},\;\lambda^{\left(2\right)}=-\lambda^{\left(1\right)},\;\lambda^{\left(3\right)}=\frac{1+i}{\sqrt{2}},\;\lambda^{\left(4\right)}=-\lambda^{\left(3\right)}$
σ=π	$\lambda^{\left(1\right)}=\frac{-1+\sqrt{2}i}{\sqrt{3}},\;\lambda^{\left(2\right)}=-\lambda^{\left(1\right)},\;\lambda^{\left(3\right)}=\frac{1+\sqrt{2}i}{\sqrt{3}},\;\lambda^{\left(4\right)}=-\lambda^{\left(3\right)}$
$\sigma =\frac{5}{4}\pi$	$\lambda^{\left(1\right)}=\frac{-1+i}{\sqrt{2}},\;\lambda^{\left(2\right)}=-\lambda^{\left(1\right)},\;\lambda^{\left(3\right)}=\frac{1+3i}{\sqrt{10}},\;\lambda^{\left(4\right)}=-\lambda^{\left(3\right)}$
$\sigma =\frac{3}{2}\pi$	$\lambda^{\left(1\right)}=i,\;\lambda^{\left(2\right)}=-\lambda^{\left(1\right)},\;\lambda^{\left(3\right)}=i,\;\lambda^{\left(4\right)}=-\lambda^{\left(3\right)}$
$\sigma =\frac{7}{4}\pi$	$\lambda^{\left(1\right)}=\frac{1+i}{\sqrt{2}},\;\lambda^{\left(2\right)}=-\lambda^{\left(1\right)},\;\lambda^{\left(3\right)}=\frac{-1+3i}{\sqrt{10}},\;\lambda^{\left(4\right)}=-\lambda^{\left(3\right)}$

**Table 4 entropy-22-00127-t004:** The parameter dependence of the eigenvalues for the complete two-phase quantum walk.

Value of σ	The Eigenvalues
σ=0	$\lambda^{\left(1\right)}=\sqrt{-i},\;\lambda^{\left(2\right)}=-\lambda^{\left(1\right)},\;\lambda^{\left(3\right)}=\sqrt{i},\;\lambda^{\left(4\right)}=-\lambda^{\left(3\right)}$
$\sigma =\frac{\pi }{4}$	$\lambda^{\left(1\right)}=\sqrt{\frac{-2\left(2-i\right)+\left(1-i\right)\sqrt{6i}}{2\{\left(1+i\right)-\sqrt{6i}\}}},\;\lambda^{\left(2\right)}=-\lambda^{\left(1\right)}$, $\lambda^{\left(3\right)}=\sqrt{\frac{-2\left(2-i\right)-\left(1-i\right)\sqrt{6i}}{2\{\left(1+i\right)+\sqrt{6i}\}}},\;\lambda^{\left(4\right)}=-\lambda^{\left(3\right)}$
$\sigma =\frac{\pi }{2}$	$\lambda^{\left(1\right)}=i,\;\lambda^{\left(2\right)}=-\lambda^{\left(1\right)},\;\lambda^{\left(3\right)}=i,\;\lambda^{\left(4\right)}=-\lambda^{\left(3\right)}$
$\sigma =\frac{3}{4}\pi$	$\lambda^{\left(1\right)}=\sqrt{\frac{-2\left(2+i\right)+\left(1+i\right)\sqrt{-6i}}{2\{\left(1-i\right)-\sqrt{-6i}\}}},\;\lambda^{\left(2\right)}=-\lambda^{\left(1\right)}$, $\lambda^{\left(3\right)}=\sqrt{\frac{-2\left(2+i\right)-\left(1+i\right)\sqrt{-6i}}{2\{\left(1-i\right)+\sqrt{-6i}\}}},\;\lambda^{\left(4\right)}=-\lambda^{\left(3\right)}$
σ=π	$\lambda^{\left(1\right)}=\sqrt{-i},\;\lambda^{\left(2\right)}=-\lambda^{\left(1\right)},\;\lambda^{\left(3\right)}=\sqrt{i},\;\lambda^{\left(4\right)}=-\lambda^{\left(3\right)}$
$\sigma =\frac{5}{4}\pi$	$\lambda^{\left(1\right)}=\sqrt{\frac{-2\left(2-i\right)+\left(1-i\right)\sqrt{6i}}{2\{\left(1+i\right)-\sqrt{6i}\}}},\;\lambda^{\left(2\right)}=-\lambda^{\left(1\right)}$, $\lambda^{\left(3\right)}=\sqrt{\frac{-2\left(2-i\right)-\left(1-i\right)\sqrt{6i}}{2\{\left(1+i\right)+\sqrt{6i}\}}},\lambda^{\left(4\right)}=-\lambda^{\left(3\right)}$
$\sigma =\frac{3}{2}\pi$	$\lambda^{\left(1\right)}=i,\;\lambda^{\left(2\right)}=-\lambda^{\left(1\right)},\;\lambda^{\left(3\right)}=i,\;\lambda^{\left(4\right)}=-\lambda^{\left(3\right)}$
$\sigma =\frac{7}{4}\pi$	$\lambda^{\left(1\right)}=\sqrt{\frac{-2\left(2+i\right)+\left(1+i\right)\sqrt{-6i}}{2\{\left(1-i\right)-\sqrt{-6i}\}}},\;\lambda^{\left(2\right)}=-\lambda^{\left(1\right)}$, $\lambda^{\left(3\right)}=\sqrt{\frac{-2\left(2+i\right)-\left(1+i\right)\sqrt{-6i}}{2\{\left(1-i\right)+\sqrt{-6i}\}}},\;\lambda^{\left(4\right)}=-\lambda^{\left(3\right)}$

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
