# Peer review of "Eigenvalues of Two-State Quantum Walks Induced by the Hadamard Walk"

_entropy, 2020, doi:10.3390/e22010127_

Round 1

Reviewer 1 Report

The present manuscript investigates the eigenvalues of discrete-time quantum walks in one-dimension. More specifically, the authors study different models and desctribe the eigenvalues for the considered models since such analysis is also related to localization effects.

According to the present manuscript form, it is not easy to follow the main results of the paper, and while reading it the impression is that a more detailed discussion of the findings is requires.

I would thus suggest the authors to perform some revisions on the manuscript, in particular:

1) At the beginning of Sec. 2, I would suggest to expand the discussion on the motivations behind the specific choice of the adopted models. Also, an introductory schematic figure on quantum walks, and the specific assumptions on its structures behind the chosen models, would be helpful to obtain a schematic overview of the paper.

2) I would suggest the authors to add a Discussion section (or to include it in the Summary) to provide some more detailed paragraphs on the results obtained in the manuscript.

Author Response

1. Page 3, just before Subsection 2:1: We added our comments on the motivations behind
the speci c choice of the adopted models.
2. We put the pattern diagrams of the QW and each QW model.
3. The Summary: We added some comments for our results.

Reviewer 2 Report

In this paper, the authors investigate the eigenvalues for some space-inhomogeneous quantum walks, using the splitted generating function method. They claim that their results are relevant for the localization of this kind of walks.

While these results appear to be scientifically sound, in my opinion the authors should make an extra effort to motivate them, and to present them in a more pedagogical way. Here is a list of the questions that should be addressed, before I can recommend this paper for publication:

- The authors should give a short introduction to the splitted generating function method for readers, like me, who are not familiar with the method. How are the numerical simulations with Mathematical related to the analytical formulae given in the text?

- They should explain the figures in more detail. What does the angular variable represent in the figures? What is the Hadamard spectrum in these figures? What implies that this part of the spectrum is excluded in the discussed models?

- The authors should discuss the consequences of the obtained results for the localization of quantum walks, since they claim that these results are relevant regarding localization.

- Model 3 in Sect. 2.3 introduces two parameters \sigma_+ and \sigma_-. What is \sigma in the formulae for the eigenvalues in this Section?

- Figures 1 and 3 are hard to see. Is there an eigenvalue movement, as appreciated in Figs. 3 and 4?

Author Response

1. Section 2-Paragraph 2: We added the short comments for the splitted generating function method(the SGF method). Also, each numerical simulation with Mathematica is the eigenvalues (given by the SGF method) movements depending on the parameter for the QW.We present in the text the eigenvalues with the model parameter for each QW model.
2. Figures: Each numerical result is the eigenvalues movements depending on the parameter for the QW. For instance, Ď• is the parameter of the unitary operator for the Wojcik model. Also, the reference for the continuous spectrum for the Hadamard walk is [17] in the manuscript, for instance. Furthermore, we added some comments for our results in the Summary.
3. Introduction-Paragraph 2, just before Subsection 2, the Summary: We added some com- ments for localization.
4. Page 8: Newly information added.
5. Figures 1 (and 2) expresses the eigenvalues movements for the two cases of the initial states. According to the SGF method, we obtain the eigenvalues for the two cases.

Round 2

Reviewer 1 Report

The authors have satisfactorily revised the manuscript according to my previous report. I can thus support its publication in Entropy.

Author Response

We would like to deeply thank to the referees for the constructive suggestions with quite carefully checking our article.

Reviewer 2 Report

This revised version certainly constitutes an improvement over the first one. In my opinion, however, the authors should make an effort to rewrite the Summary. In particular:

- "According to the previous studies [8–10,12,13], our QW models have the eigenvalues for the eigenfunctions in l^2 -space on Z, i.e., localization occur, ...".

What is the connection between eigenvalues and localization?

- "Specifically, we revealed that the eigenvalues do not emerge on the region of the continuous spectrum of the Hadamard walk, which implies that the continuous spectrum of our QW models may coincide with that of the Hadamard walk ...".

This sentence is confusing, and the authors should explain in more detail what they mean.

Author Response

Please see in the attachment.
